# Genetic Diversity of Hepatitis B and C Viruses Revealed by Continuous Surveillance from 2015 to 2021 in Gabon, Central Africa

**DOI:** 10.3390/microorganisms11082046

**Published:** 2023-08-09

**Authors:** Haruka Abe, Yuri Ushijima, Rodrigue Bikangui, Georgelin Nguema Ondo, Christelle M. Pemba, Vahid R. Zadeh, Patrick I. Mpingabo, Hayato Ueda, Selidji T. Agnandji, Bertrand Lell, Jiro Yasuda

**Affiliations:** 1Department of Emerging Infectious Diseases, National Research Center for the Control and Prevention of Infectious Diseases (CCPID), Nagasaki University, Nagasaki 852-8523, Japan; abeh@nagasaki-u.ac.jp (H.A.); christellepemba@nagasaki-u.ac.jp (C.M.P.); vahid.r.kashani@gmail.com (V.R.Z.); patrickmpingabo@gmail.com (P.I.M.); bb20120022@ms.nagasaki-u.ac.jp (H.U.); 2Department of Emerging Infectious Diseases, Institute of Tropical Medicine (NEKKEN), Nagasaki University, Nagasaki 852-8523, Japan; ushijima-yu@md.tsukuba.ac.jp; 3Vietnam Research Station, Institute of Tropical Medicine (NEKKEN), Nagasaki University, Nagasaki 852-8523, Japan; 4Division of Biomedical Science, Institute of Medicine, University of Tsukuba, Tsukuba 305-8577, Japan; 5Centre de Recherches Médicales de Lambaréné (CERMEL), Lambaréné BP. 242, Gabon; bikrod16@gmail.com (R.B.); georgelinkluivert9@gmail.com (G.N.O.); agnandjis@cermel.org (S.T.A.); bertrand.lell@cermel.org (B.L.); 6Institute for Tropical Medicine, University of Tübingen, 72074 Tübingen, Germany; 7Division of Infectious Diseases and Tropical Medicine, Medical University of Vienna, 1090 Vienna, Austria; 8Graduate School of Biomedical Sciences, Nagasaki University, Nagasaki 852-8523, Japan

**Keywords:** hepatitis virus, HBV, HCV, surveillance, Gabon, Africa

## Abstract

Viral hepatitis remains one of the largest public health concerns worldwide. Especially in Central Africa, information on hepatitis virus infections has been limited, although the prevalence in this region has been reported to be higher than the global average. To reveal the current status of hepatitis B and C virus (HBV and HCV) infections and the genetic diversity of the viruses, we conducted longitudinal surveillance in Gabon. We detected 22 HBV and 9 HCV infections in 2047 patients with febrile illness. Genetic analyses of HBV identified subgenotype A1 for the first time in Gabon and an insertion generating a frameshift to create an X-preC/C fusion protein. We also revealed that most of the detected HCVs belonged to the “Gabon-specific” HCV subtype 4e (HCV-4e), and the entire nucleotide sequence of the HCV-4e polyprotein was determined to establish the first reference sequence. The HCV-4e strains possessed resistance-associated substitutions similar to those of other HCV-4 strains, indicating that the use of direct-acting antiviral therapy may be complex. These results provide a better understanding of the current situation of hepatitis B and C virus infections in Central Africa and will help public health organizations develop effective countermeasures to eliminate chronic viral hepatitis in this region.

## 1. Introduction

Chronic hepatitis virus infections are the major global cause of chronic liver disease, cirrhosis, and liver cancer. Hepatitis B virus (HBV) infection is a public health concern worldwide, with an estimated 296 million people chronically positive for the HBV surface antigen (HBsAg) [1]. Although the global all-age prevalence of chronic HBV infection decreased from 6.0% in 1990 to 4.1% in 2019, the number of HBV-related deaths increased from 555,000 to 820,000 worldwide between 1990 and 2019 [2,3]. In particular, the prevalence of chronic HBV infections has remarkably declined in children under the age of 5 years owing to the introduction of effective vaccines [2]. Hepatitis C virus (HCV) infection also remains an important health threat, with 56.8 million chronically infected people, resulting in 290,000 deaths yearly [3]. It was estimated that viral hepatitis B- and C-related diseases, in 2020, were responsible for a significantly higher number of deaths (1.1 million deaths) than those caused by HIV infection and malaria (0.68 and 0.63 million deaths, respectively) and were equivalent to the number of deaths caused by tuberculosis (1.3 million deaths) [3]. Particularly in sub-Saharan Africa, the prevalence of chronic infections with either HBV or HCV has been much higher than the global average [4], possibly due to difficulties with mass vaccination and the lack of knowledge to prevent infections in communities.

HBV and HCV are genetically divergent, probably due to the easy acquisition of genomic mutations with an error-prone viral polymerase, enabling the viruses to survive in the presence of human innate and acquired immune responses [5]. HBV is highly divergent and has so far been categorized into nine confirmed (A–I) and one proposed (J) genotypes, and numerous subgenotypes can be further classified [6]. Several genotypes and/or mutations have been reported to show strong relationships with clinical severity [7,8,9], indicating that the accurate genotyping of HBV is essential for a better understanding of the condition of patients. Sub-Saharan Africa is one of the regions where the prevalence of chronic hepatitis B is much higher than the global average, although a limited number of surveillance studies have been conducted [4,10]. In this geographic region, several sporadic reports have shown that genotypes A and E predominate. For genotype A, sub-genotype A1 (HBV-A1) is widespread across the African continent, and HBV-A3, A4, A5, and A7 have been reported mainly in West and Central Africa [11,12]. HBV genotype E (HBV-E), which is related to the precore stop codon mutation G1896A, is endemic to West Africa and several Central African countries [11,13].

HCV is also highly divergent, consisting of eight genotypes (genotypes 1–8), and each genotype is further divided into numerous subtypes [14,15,16,17]. Although interferon-free therapeutic regimens with direct-acting antivirals (DAAs) provide remarkable efficacy against chronic HCV infections, genotype-specific and sporadic resistance-associated substitutions (RASs) have been identified through a sequence analysis of HCV in patients with DAA failure [18]. The patterns of RASs differ depending on the genotype and subtype, indicating the importance of HCV genotyping in predicting patients’ susceptibility to DAAs [18,19,20]. A large number of genetic surveillance studies of HCV infections have clarified the genotypes/subtypes prevalent in Europe, the Americas, Asia, and Oceania. Particularly in sub-Saharan Africa, limited data are available to understand prevalent genotypes/subtypes and RASs [21].

Gabon, located in Central Africa, is one of the countries where HBV and HCV are highly endemic [3,22]. A previous sero-surveillance study showed a high prevalence of HBsAg in urban (12.9%) and rural (7.6%) populations and in pregnant women (9.2%) in Gabon [23,24]. Similarly, seropositivity for HCV was much higher in Gabon (11.2%) than the global average (0.8%) [3,25]. Gabonese HBV strains have been reported to belong to subtypes A and E, especially A3, which has been found mainly in Central Africa [23,24]. In addition, previous studies have reported that most HCV strains in Gabon belong to subtype 4e (HCV-4e) [26]. Interestingly, although only partial NS5B sequences are available, HCV-4e seems to be confined to Gabon, and there has been no report of the detection of this subtype outside this country [21]. However, large-scale genetic surveillance is needed to understand the current trend of genotypes and mutations that may affect the severity and efficacy of DAA therapy, since past studies were conducted at a relatively small scale [25,26].

In this study, we conducted longitudinal genetic surveillance for HBV and HCV, which are public health concerns in Gabon. We examined the demographic characteristics of patients with each viral infection and identified several mutations that potentially affect viral characteristics or their resistance to DAA. Moreover, we determined the entire nucleotide sequence of the polyprotein open reading frame (ORF) of HCV-4e. The findings of the current study provide new insights into the high diversity of HBV and HCV in Central Africa.

## 2. Materials and Methods

### 2.1. Sample Collection

A total of 2179 patients with fever (body temperature ≥ 37.5 °C) who visited the Centre de Recherches Médicales de Lambaréné (CERMEL) and the Albert Schweitzer Hospital in Lambaréné were recruited in a surveillance study between May 2015 and July 2021. In the present study, the age of the participants was restricted to ≥1 year. Demographic information (age and sex at birth) of the recruited participants was also collected.

### 2.2. Ethics Statement

This study was approved by the institutional review boards of CERMEL and Nagasaki University (approval numbers CEI-007 and 170921177, respectively). Written informed consent was obtained from all participants or their parents.

### 2.3. Viral Genome Extraction and Reverse Transcription Quantitative PCR (RT-qPCR)

The viral genome was extracted from 140 μL of each serum sample using a QIAamp Viral RNA Mini Kit (Qiagen, Hilden, Germany) according to the manufacturer’s instructions. RT-qPCR was performed in a 20 μL reaction using One Step PrimeScript III RT-qPCR Mix (Takara Bio, Shiga, Japan) as described previously [27]. For the detection of HBV genomes, the reaction mixture contained 10 μL 2× One Step PrimeScript RT-qPCR Mix, 0.5 μM of each primer, 0.25 μM of TaqMan probe, 0.4 μL of ROX Reference Dye, 2 μL of RNA template, and RNase-free water to total 20 μL. To detect HCV genomes, the concentration of each primer was changed to 1.0 μM. The sequences of the primers and probes have been reported previously [28,29]. RT-qPCR assays were carried out with a StepOnePlus instrument (Thermo Fisher Scientific, Waltham, MA, USA) under the following conditions: 5 min at 52 °C, 10 s at 95 °C, and 45 cycles of 5 s at 95 °C and 35 s at 60 °C, as described previously [30]. Data collected from RT-qPCR assays were analyzed using the software included in the StepOnePlus system. Samples reaching threshold cycle (Ct) values < 40 were considered positive. RT-qPCR assays were performed in duplicate for virus-positive samples to confirm positivity.

### 2.4. Genotyping of Hepatitis Virus Strains

Nested PCR was performed to amplify the PreS1-PreS2-S region (1180 nucleotides (nt)) of the HBV genome with PrimeSTAR GXL DNA Polymerase (Takara Bio) using previously reported primers [31]. The first PCR reaction mixture contained 5 μL of 5× PrimeSTAR GXL Buffer, 0.4 μM of outer primers HBPr1 and HBPr135, 2 μL of dNTP Mixture (2.5 μM each), 0.5 μL of PrimeSTAR GXL DNA Polymerase, 2 μL of RNA template, and RNase-free water to total 25 μL. The nested PCR reaction mixture contained 0.4 μM of inner primers HBPr2 and HBPr94 and 1 μL of first PCR product, and the rest of the components were the same as those in the first PCR mix. According to the manufacturer’s instructions, PCR was performed under the following conditions: 35 cycles of 10 s at 98 °C, 15 s at 55 °C, and 1 min at 68 °C.

For the amplification of the partial NS5B region of HCV, primers were redesigned using a previously reported primer sequence [32] to efficiently amplify the NS5B gene of HCV-4, which was reported to be highly prevalent in Central Africa [21,26]. RT-qPCR was performed to amplify the partial NS5B region (385 nt) of the HCV genome using a PrimeScript II High Fidelity One Step RT-PCR Kit (Takara Bio). The RT-qPCR reaction mixture contained 12.5 μL of 2× One Step High Fidelity Buffer, 0.4 μM of primers, 0.5 μL of PrimeScript II RT Enzyme Mix, 2 μL of PrimeSTAR GXL DNA Polymerase, 1 μL of RNA template, and RNase-free water to total 25 μL. According to the manufacturer’s instructions, PCR was performed under the following conditions: 10 min at 45 °C and 2 min at 45 °C, followed by 40 cycles of 10 s at 98 °C, 15 s at 55 °C, and 30 s at 68 °C.

After agarose gel purification with a QIAquick Gel Extraction Kit (Qiagen, Hilden, Germany), the PCR products were processed using a BigDye Terminator v3.1 Cycle Sequencing Kit (Thermo Fisher Scientific) and analyzed with a 3500 genetic analyzer (Thermo Fisher Scientific) to obtain sequence data. Sequenced fragments were assembled using CLC Main Workbench software v22 (Qiagen), and consensus sequences were extracted. The genotypes of the virus strains detected in this study were determined by BLAST analysis (https://blast.ncbi.nlm.nih.gov/Blast.cgi, accessed on 5 April 2022).

### 2.5. Whole-Genome Sequencing

Four fragments covering the complete genome of HBV were amplified via PCR using the primers shown in Appendix A. The PCR products were purified from agarose gels and subjected to sequencing using the same method as that used for genotyping. Sequence data were assembled into a complete HBV genome using CLC Main Workbench software (Qiagen).

To obtain the whole ORF sequence of the HCV polyprotein, libraries were prepared from extracted viral RNA samples using a NEBNext Ultra II RNA Library Prep Kit (New England Biolabs, Ipswich, MA, USA) according to the manufacturer’s instructions. After quality and quantity checks of each library using an Agilent 2100 Bioanalyzer (Agilent Technologies, Santa Clara, CA, USA) with a High Sensitivity DNA Kit (Agilent), sequencing was performed using a 300-cycle MiSeq Reagent Kit v2 (Illumina, San Diego, CA, USA) on a MiSeq sequencer (Illumina). Similar to a previous report, mapping of the paired-end reads was performed on CLC Genomics Workbench software (Qiagen) using the whole-genome sequence of the genotype 4 strain as a template [27]. Consensus sequences were extracted and aligned with reference strains using BioEdit 7.0.5.3 software (http://www.mbio.ncsu.edu/BioEdit/bioedit.html, accessed on 5 April 2022). To identify RASs in HCV-4 strains, we selected one strain from each HCV-4 subtype with criteria that the selected strain had the smallest number of ambiguous nucleotides and a longer sequence than others within each subtype.

### 2.6. Phylogenetic Analysis

To infer the phylogenetic relationships of the HBV and HCV strains detected in this study, phylogenetic analyses were performed with representative sequences of the HBV and HCV strains reported in Gabon, as reference, using MEGA 7 software (https://www.megasoftware.net/, accessed on 5 April 2022). The analyses were performed to infer neighbor-joining trees using a model of the number of differences and gamma distribution of the substitution rate. A total of 5000 bootstrap replicates were generated. To compare with data from countries outside sub-Saharan Africa, HBV reference strains were widely selected from each continent and from various time points during the collection period. For the phylogenetic analysis of complete HCV genome sequences, representative HCV genotype 4 sequences were obtained from GenBank, aligned, and checked manually for gaps to remove. For better visualization, the phylogenetic trees were modified using FigTree v1.4.2 software (http://tree.bio.ed.ac.uk/software/figtree, accessed on 5 April 2022).

### 2.7. Statistical Analysis

Statistical data analysis was performed using GraphPad Prism 8 software (GraphPad Software, San Diego, CA, USA). Fisher’s exact tests were used to determine significant differences in comparisons of general categorical variables. Results were considered statistically significant with *p*-values < 0.05.

### 2.8. Sequence Data Availability

The HBV and HCV strain sequences obtained in this study were deposited in GenBank under the accession numbers LC773618–LC773640.

## 3. Results

### 3.1. HBV and HCV Detection via RT-qPCR and Demographic Information of Patients with HBV or HCV Infections

We collected serum samples from 2179 patients with fevers as part of a project to investigate the infectious diseases prevalent in Gabon. We selected 2047 samples that had enough demographic information to conduct an RT-qPCR and epidemiological analyses. The RT-qPCR screening identified 22 HBV-positive and 9 HCV-positive patients. HBV was detected in patients aged 2–80 years, whereas the age of HCV-positive patients was relatively high, ranging from 18 to 81 years. Comparing the age and male/female ratios between the virus-positive and virus-negative populations, we determined that the virus-positive population was significantly older than the virus-negative population for both HBV and HCV, although the male/female ratios were not significantly different (Table 1).

### 3.2. Phylogenetic Analysis and Genetic Diversity of the HBV Strains Detected in Gabon

To infer the phylogeny and clarify the genetic diversity of the HBV strains detected in this study, we performed partial S gene amplification via a PCR from the HBV-positive samples and determined their sequences. The phylogenetic tree showed that the current HBV strains belonged to subgenotypes A1 and A3 (HBV-A1 and HBV-A3) and genotype E (HBV-E) (Figure 1). Notably, this is the first report of HBV-A1 in Gabon. Then, we searched the genetically closest HBV strains to the sequences obtained in this study using BLAST. The HBV-A3 strains detected in this study were highly related to previously reported HBV-A3 strains in Gabon, whereas the HBV-A1 strain from this study was genetically close to a South African HBV-A1 strain detected in 2013 (Appendix A). Additionally, the HBV-E strain detected in this study was close to an isolate from Guinea, indicating the wide distribution of African HBV-E in West and Central Africa (Appendix A).

### 3.3. Unusual Fusion of the HBV X (HBx) and PreC/C Proteins Caused by Rare Insertions in HBV Genomes

The whole-genome sequencing analysis of the identified HBV strains showed that the HBV-A3 strain SYMAV-D0232 possessed a seven-nucleotide insertion at position 1826 which caused a frameshift (Figure 2a). As a result, the stop codon of HBx disappeared, and the HBx and preC/C proteins were fused in frame, creating a novel fusion protein (Figure 2b). We previously reported a similar mutation creating an HBx-preC/C fusion protein in strain SYMAV-L0084-#1-B (Figure 2a,b) [28]. Under these frameshift conditions, the amino acid sequence of the preC/C protein of strain SYMAV-D0232 changed immediately after the start codon, and stop codons appeared at position 35 in the mutated preC/C protein (Figure 2b).

### 3.4. Phylogenetic Analysis of the HCV Strains Detected in Gabon

The phylogenetic analyses indicated that most of the HCV strains detected in this study belonged to the Gabon-specific HCV-4e cluster, together with previously reported Gabonese HCV-4e strains (Figure 3). HCV-4f and HCV-4k strains were also identified in this study, clustering with previously reported Gabonese strains correspondingly (Figure 3).

Since only NS5B gene sequences were available for the Gabon-specific HCV-4e strains, we attempted to obtain whole ORF sequences to establish reference sequences for this group. We successfully obtained the complete ORF sequences of the HCV-4e, HCV-4f, and HCV-4k polyproteins. The phylogenetic tree inferred using whole ORF polyprotein sequences clearly revealed that HCV-4e was genetically close to HCV-4c and HCV-4a (Figure 4). The phylogenetic analyses’ results were consistent with those for nucleotide identity between HCV-4e and other HCV-4 subtypes (Figure 5); for example, HCV-4c was the closest subtype to HCV-4e in both phylogenetic and nucleotide identity (81.95%) analyses (Figure 4 and Figure 5).

### 3.5. Potential DAA Resistance of HCV-4e Strains

To date, several RASs have been identified in the ORFs of DAA-targeting viral proteins, raising concerns about therapeutic regimens. We checked previously identified RASs in HCV-4 subtypes, including the newly sequenced HCV-4e (Table 2). HCV-4e possessed protein RASs similar to those of the other HCV-4 strains: V36L and S122T in NS3; M28L, Q30R, L31M, and H58P in NS5A; and S556G in NS5B (Table 2). Several other HCV-4 subtypes possess additional RASs in their proteins, such as E62S, Y93R, C316H, and V321I in the NS5B protein in HCV-4g.

## 4. Discussion

Chronic hepatitis virus infections constitute one of the most serious public health concerns worldwide. In 2016, the World Health Assembly adopted the WHO Global Health Sector Strategy on Viral Hepatitis (WHO-GHSS)’s goal of eliminating viral hepatitis as a public health threat. The WHO-GHSS suggested decreasing the number of new hepatitis B infections from 1.5 million in 2020 to 170,000 by 2030 and the number of new hepatitis C infections from 1.575 million in 2020 to 350,000 by 2030 [33]. The WHO-GHSS also suggested an increase in the number of countries validated for the elimination of hepatitis B and/or hepatitis C to up to 20 countries in 2030 [33]. Moreover, the UN described combating hepatitis in Goal 3.3 of the UN Sustainable Development Goals [34].

Sub-Saharan Africa is one of the regions where viral hepatitis is expected to be highly prevalent, although information on these kind of infections has been limited in the region. Particularly in Gabon, Central Africa, HBV prevalence is estimated to be more than 6.7% [1,2]. In such high-prevalence regions, hepatitis viruses show great diversity, and novel subgenotypes/subtypes and RASs are frequently found. In the current study, we showed further evidence of divergence among HBV strains in Gabon and detected HBV-A1 for the first time, as well as the previously reported HBV-A3, HBV-A5, and HBV-E strains [23,24,28]. Interestingly, we previously detected two distinct HBV strains potentially generated by the ongoing evolution of HBV in an infected patient [28]. The HBV strain SYMAV-L0084-1B showed a rare one-nucleotide insertion around nucleotide position 1826, creating a fusion protein between proteins HBx and preC/C [28]. In this study, we identified a rare seven-nucleotide insertion at the same position in the HBV genome, indicating a possible unique pattern of HBV evolution in Gabon. Due to the protein processing of the preC/C precursor protein to generate the mature HBeAg protein, the creation of the HBx-preC fusion protein may have a few effects on the HBeAg function. However, the function of HBx fused with the preC N-terminus has not been well analyzed, so we should continuously monitor the prevalence of these “evolving” HBV strains and the severity of patients who are infected with HBVs with rare insertions in Gabon.

In this study, we analyzed the genetic diversity of HBV and HCV in samples collected in Gabon, although previous serological studies have reported a much higher prevalence of HBV and HCV in Gabon, more than 7% and 10%, respectively [23,24,25]. According to a previous report from Gabon, positivity for anti-HBV-core IgG and anti-HCV IgG antibodies clearly increased with age [25,35]. Whereas, the mean age of our samples was approximately 12.9 years (Table 1), indicating a possibility that the sampling population was relatively young and might show a lower prevalence rate for HBV and HCV. In addition, the HCV RNA was detected via a nested RT-PCR in samples fewer than half the number of ELISA-positive ones in the previous study in Gabon [26]. Our surveillance was based on a qPCR but not on antibody or antigen detection, possibly generating a difference in prevalence between previous studies and the current study.

Hong et al. recently showed that the N-terminal signal peptide of the immature precore protein (p25) translated from the precore mRNA should be mostly removed by the signal peptidase in the ER lumen at the amino acid position of −11/−10, leading to the production of HBeAg after further processing at its C-terminal domain [36]. This means that HBeAg might be normal even in the fusion protein due to the post-translational cleavage of HBeAg from the fusion protein, whereas HBx possibly possessed several extra amino acid residues corresponding to −25 and −11 of the precore protein at its C-terminus even after the cleavage of HBeAg from the fusion protein. Prieto et al. and Hernández et al. recently showed that HBx localization was unaffected even though they fused GFP to the C-terminus of HBx [37,38]. Therefore, several extra residues at the C-terminus would lead to a minimal effect on the HBx function. As the function of HBx is essential to initiating and maintaining HBV replication after infection, HBx would function appropriately after being processed from the fusion protein in the sample D0232 [39].

Since DAAs have been recognized as highly effective therapeutics against HCV infections, the number of deaths due to hepatitis C has been reduced [3,17,18]. However, DAA-resistant HCVs possessing RASs have emerged, hindering the effectiveness of therapeutic regimens [18,19,20]. HCV diversity is expected to be very high in sub-Saharan Africa because HCVs have prevailed and evolved in this region for a long time [40]. A number of HCV-4 subtypes that show several RASs are prevalent in Central Africa, whereas HCV-4e has been mostly confined to Gabon, although the whole genome sequence is still unavailable [21,25,26]. In this study, we detected HCV-4e strains and determined the complete nucleotide sequence of the HCV-4e polyprotein for the first time. The whole nucleotide sequence of the HCV-4e polyprotein ORF was genetically closest to that of HCV-4c, which has also been detected in Gabon. 

As reported previously, entire ORF sequences are required to establish confirmed subtypes and infer phylogeny more appropriately than with the sequence of the NS5B region, although the NS5B sequence is sufficient to determine the genotypes/subtypes of the detected strains [15,16]. HCV-4e showed the same RASs that have been observed in other HCV-4 strains, indicating that HCV-4e and other HCV-4 subtypes may have similar resistance abilities against DAAs. As emphasized elsewhere, clinical studies using DAAs as well as HCV surveillance have been very limited in sub-Saharan Africa [21]. Further efforts should be made to understand the characteristics of HCV prevalence in this region, including that of Gabon-specific HCV-4e.

In Table 2, the positions Q30 and E62 in the NS5A region were divergent within the HCV-4 strains. Relating to these divergent substitutions, there are several previous sophisticated studies that investigated the efficacy of DAA against a wide range of HCV genotypes from genotype 1 to 7 [41,42,43]. These studies revealed that amino acid residues, Q30, L31, and Y93, in the NS5A region were highly divergent among genotypes, and that the specific patterns of the substitutions of these residues were associated with high-level resistance to DAA, such as Q30H/R/S, L31M/F, and Y93H/C/N/S. The E62 position was also divergent among genotypes, although substitutions in this position were weakly associated with resistance [42]. Thus, the amino acid residues highly related with resistance were known to be divergent not only within genotypes but also among genotypes. In particular, a number of HCV-4 strains possessed both Q30R/S and L31M substitutions that might cause high-level resistance. It is required to validate the DAA efficacy in Central Africa, where HCV-4 strains are highly prevalent.

The limitation of this study is that our analysis was based on passive surveillance. qPCR tests for HBV and HCV were used for patients who visited local hospitals in Gabon, which might induce a bias in the selection of the target population. Actually, the mean age of all participants was 12.9 years, indicating that a large number of participants were children less than 12 years old. Moreover, previous studies conducted in Gabon were serological screenings for HBV and HCV, and PCR testing was used only for antibody/antigen-positive samples, although our surveillance performed qPCRs to detect HBV and HCV [23,24,25,26]. To better understand their prevalence, it might be necessary to combine qPCRs and antibody/antigen detection in HBV and HCV surveillance.

To clarify the current status of viral diseases, we have conducted continuous surveillance in Gabon, targeting dengue virus, chikungunya virus, Zika virus, hepatitis A virus, West Nile virus, and Severe Acute Respiratory Syndrome Coronavirus 2 (SARS-CoV-2) [27,30,44,45,46]. Through all these studies, more viruses were identified in Gabon than expected, indicating the limited information available on viral diseases in Central Africa. The current study revealed the prevalence and diversity of HBV and HCV in Central Africa, with a possible contribution to public health in this region. However, in order to maximize these results, it may be necessary to conduct clinical studies using effective vaccines and antivirals for the prevention and control of infectious diseases and to monitor the number of cases to detect an outbreak. Importantly, in Egypt, there have been large-scale clinical studies on hepatitis C to evaluate DAA efficacy against HCV subtypes prevalent in this African country’s population [47,48,49]. The results would greatly contribute to establishing effective therapeutic regimens. Central African countries require continuous support to conduct infectious disease clinical studies that may help eliminate viral hepatitis from this region.

## Figures and Tables

**Figure 1 microorganisms-11-02046-f001:**
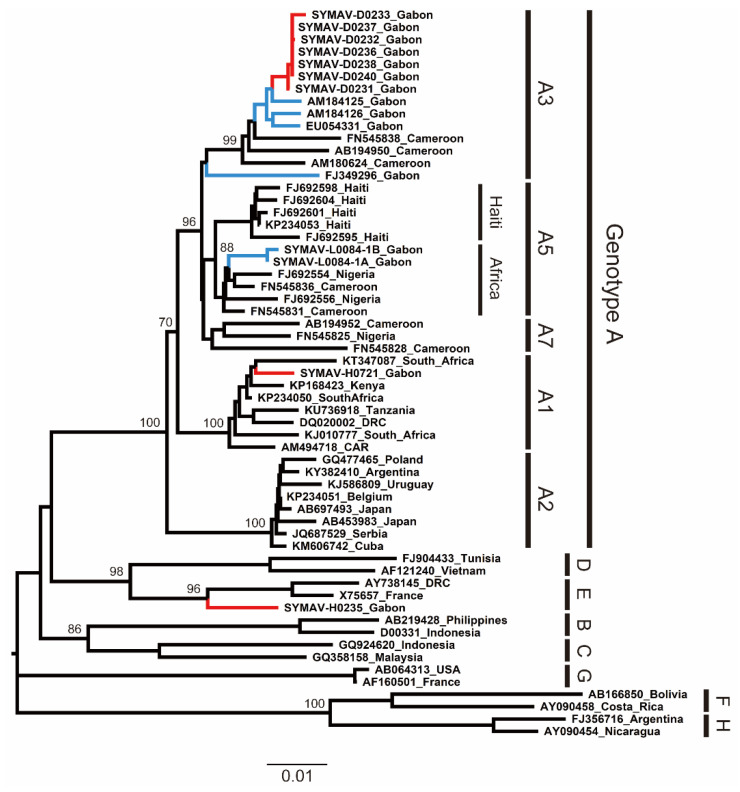
Phylogenetic analysis of HBV strains detected in Gabon using preS region (1180 nt) sequences. Genotypes and subgenotypes are indicated on the right. In red are the HBV strains from this study; in blue are the HBV strains reported for Gabon in past studies. Bootstrap values ≥ 70% are shown at the corresponding nodes. Scale bar: nucleotide substitutions per site.

**Figure 2 microorganisms-11-02046-f002:**
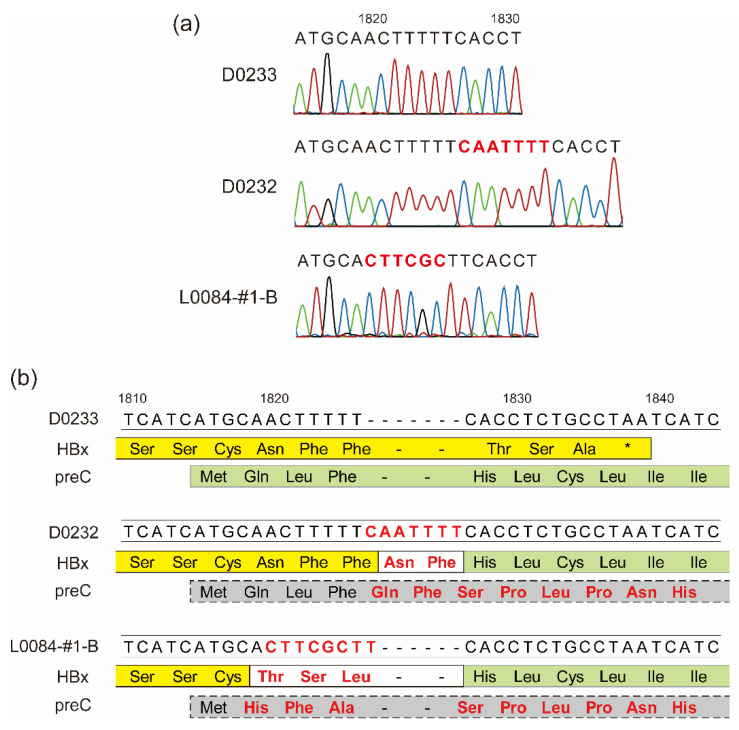
Frameshift-generating insertions expected to create fusion proteins. (**a**) Insertions and mutations detected in some HBV strains in Gabon. The nucleotide sequence of strain D0233 is shown for reference. Representative chromatograms for the HBV strains are shown under each nucleotide sequence. Colors in chromatograms depict each nucleotide. Nucleotide changes and insertions are shown in red. (**b**) The predicted amino acid sequences of the end and beginning of the HBx and preC/C ORFs, respectively, from nucleotides 1810–1840 are shown below their nucleotide sequences. In yellow: amino acid sequence of HBx; in green: amino acid sequence of preC; in gray: amino acid sequence of the mutated preC/C. Nucleotide changes and insertions are shown in red as in (**a**). New or substituted amino acid residues are also presented in red. *: stop codon.

**Figure 3 microorganisms-11-02046-f003:**
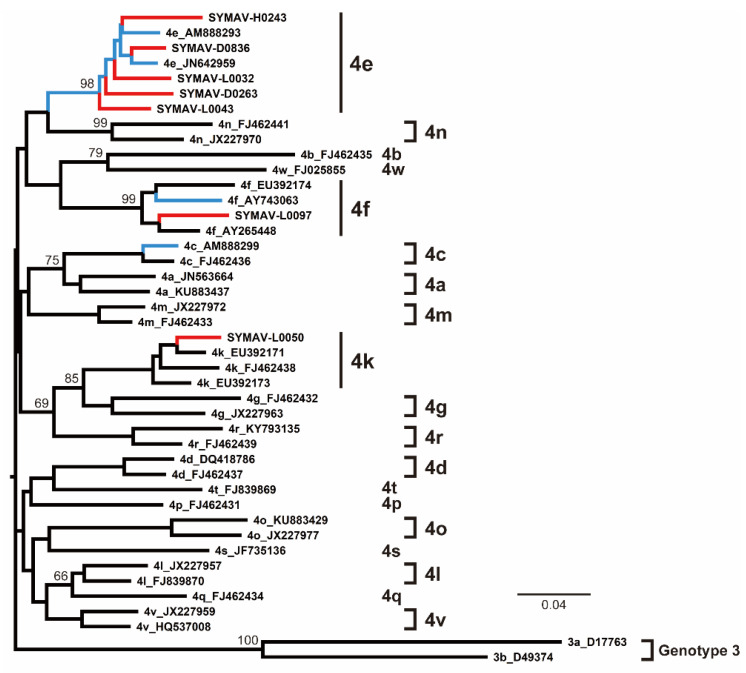
Phylogenetic analysis of HCV strains from Gabon using the NS5B region (385 nt). Genotypes and subtypes are indicated on the right. In red are the HCV strains from this study; in blue are the HCV strains reported for Gabon in past studies. Bootstrap values ≥ 60% are shown at the corresponding nodes. Scale bar: nucleotide substitutions per site.

**Figure 4 microorganisms-11-02046-f004:**
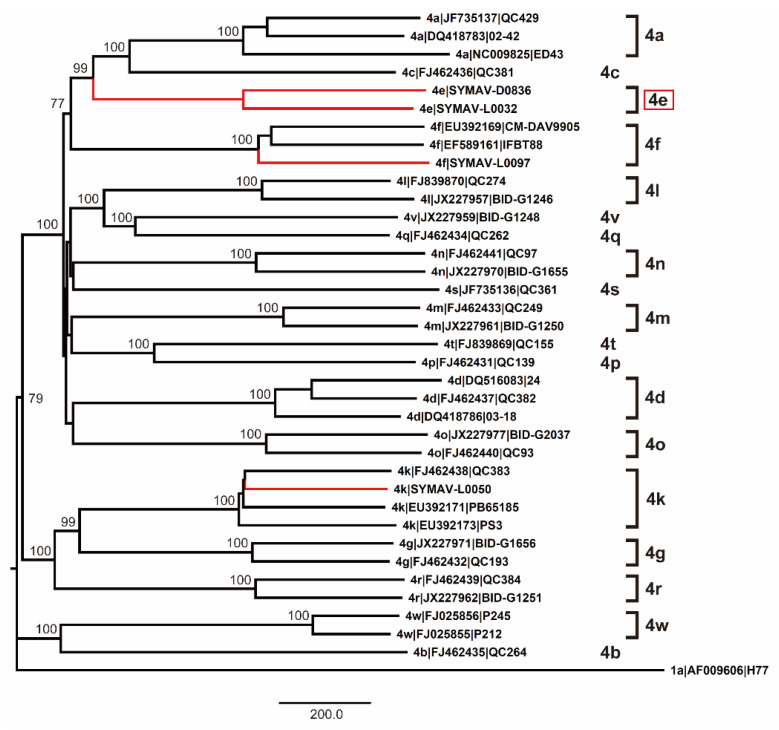
Phylogenetic analysis of HCV-4 strains using complete polyprotein ORF nucleotide sequences. Genotypes and subtypes are indicated on the right. In red: strains whose polyprotein ORFs were sequenced in this study. Bootstrap values ≥ 70% are shown at the corresponding nodes. Scale bar: nucleotide substitutions per site.

**Figure 5 microorganisms-11-02046-f005:**
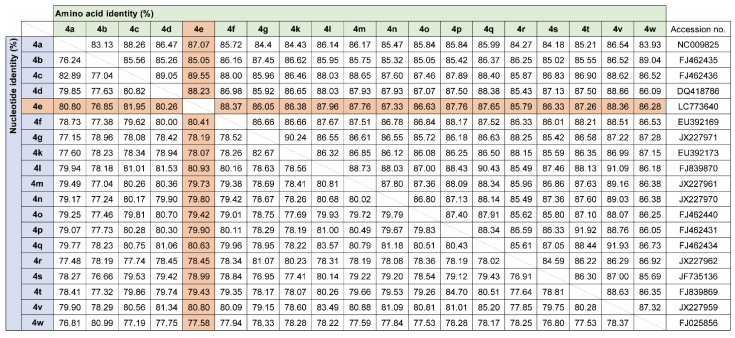
Nucleotide and amino acid identity among HCV-4 polyprotein ORFs.

**Table 1 microorganisms-11-02046-t001:** Comparison of epidemiological information between HBV- and HCV-positive participants.

	RT-qPCR-Positive Patients		RT-qPCR-Negative Patients		*p*-Value	
Virus	Age (95% CI)	M/F Ratio	Age (95% CI)	M/F Ratio	Age	M/F Ratio
HBV	21.55 (13.70–29.39)	0.45	12.97 (12.28–13.66)	0.48	<0.05	0.83
HCV	48.56 (32.61–64.50)	0.33	12.90 (12.23–13.58)	0.48	<0.001	0.38

**Table 2 microorganisms-11-02046-t002:** Resistance-associated substitutions for HCV-4 genotypes.

Protein	Position	4a	4b	4c	4d	4e	4f	4g	4k	4l	4m	4n	4o	4p	4q	4r	4s	4t	4v	4w
NS3	V36	L	L	L	L	L	L	L	L	L	L	L	L	L	L	L	L	L	L	L
	Q41	-	-	-	-	-	-	-	-	-	-	-	-	-	-	-	-	-	-	-
	F43	-	-	-	-	-	-	-	-	-	-	-	-	-	-	-	-	-	-	-
	T54	-	-	-	-	-	-	-	-	-	-	-	-	-	-	-	-	-	-	-
	V55	-	-	-	-	-	-	-	-	-	-	-	-	-	-	-	-	-	-	-
	Y56	-	-	-	-	-	-	-	-	-	-	-	-	-	-	-	-	-	-	-
	N77	-	-	-	-	-	-	-	-	-	-	-	-	-	-	-	-	-	-	-
	Q80	-	-	-	-	-	-	-	-	-	-	-	-	-	-	-	-	-	-	-
	S122	T	-	T	T	T	T	T	T	T	-	T	T	T	N	T	N	T	N	T
	R155	-	-	-	-	-	-	-	-	-	-	-	-	-	-	-	-	-	-	-
	A156	-	-	-	-	-	-	-	-	-	-	-	-	-	-	-	-	-	-	-
	D168	-	-	-	-	-	-	-	-	-	-	-	-	-	-	-	T	-	-	-
	I/V170	-	-	-	-	-	-	-	-	-	-	-	-	-	-	-	-	-	-	-
NS5A	L23	-	-	-	-	-	-	-	-	-	-	-	-	-	-	-	-	-	-	-
	K24	-	-	-	-	-	-	-	-	-	-	-	-	-	-	-	-	-	-	-
	M28	V	L	L	L	L	L	L	L	L	L	L	-	L	L	-	L	L	L	-
	Q30	L	S	R	R	R	R	L	R	R	S	R	T	R	R	R	R	R	R	S
	L31	M	M	M	M	M	M	M	-	M	M	M	M	M	M	-	-	M	M	M
	P32	-	-	-	-	-	-	-	-	-	-	-	-	-	-	-	-	-	-	-
	S38	-	-	-	-	-	-	-	-	-	-	-	-	-	-	-	-	-	-	-
	H54	-	-	-	-	-	-	-	-	-	-	-	-	-	-	-	-	-	-	-
	H58	P	P	P	T	P	P	P	P	P	R	T	P	P	P	P	P	P	P	P
	E62	D	-	Q	-	-	-	S	-	-	R	-	-	-	-	S	-	-	-	-
	A92	-	T	-	-	-	-	-	-	-	-	-	-	-	-	-	-	-	-	-
	Y93	-	H	-	-	-	-	R	-	-	-	-	-	-	-	-	-	-	-	S
NS5B	S96	-	-	-	-	-	-	-	-	-	-	-	-	-	-	-	-	-	-	-
	N142	-	-	-	-	-	-	-	-	-	-	-	-	S	-	-	-	-	-	S
	L159	-	-	-	-	-	-	-	-	-	-	-	-	-	-	-	-	-	-	-
	S282	T	-	-	-	-	-	-	-	-	-	-	-	-	-	-	-	-	-	-
	C/F/M289	-	-	-	-	-	-	-	-	-	-	-	-	-	L	-	-	-	L	-
	L320	-	-	-	-	-	-	-	-	-	-	-	-	-	-	-	-	-	-	-
	C316	-	-	-	-	-	N	H	-	-	-	-	-	-	-	H	-	-	-	-
	V321	-	-	-	-	-	-	I	-	-	-	-	-	-	-	I	-	-	-	-
	S556	G	G	G	G	G	G	G	G	G	G	G	G	G	G	N	G	G	G	G

## Data Availability

Genomic data of the newly sequenced samples were deposited in the GenBank database with accession numbers LC773618–LC773640.

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
