# Peer review of "Genetic Diversity of Hepatitis B and C Viruses Revealed by Continuous Surveillance from 2015 to 2021 in Gabon, Central Africa"

_microorganisms, 2023, doi:10.3390/microorganisms11082046_

Round 1

Reviewer 1 Report

In this manuscript, the authors investigated the current status of hepatitis B and C viruses (HBV and HCV) infections and the genetic diversity of the viruses in Gabon, Central Africa. 22 HBV and 9 HCV infections detected in 2,047 patients with febrile illness. The authors found that the subgenotype A1 HBV was identified for the first time in Gabon with an insertion generating a frameshift to create an X-preC/C fusion protein, that most of the detected HCVs belonged to the “Gabon-specific” HCV subtype 4e, and that the HCV-4e strains possessed resistance-associated substitutions similar to those of other HCV-4 strains. So The authors concluded that the results of this study provide a better understanding of the current situation of HBV and HCV infections in Central Africa, and will help public health organizations to develop effective countermeasures for elimination of chronic viral hepatitis in this region. 

This is a longitudinal surveillance to investigate the current status of HBV and HCV infections in Gabon. The study was well conducted, and the results was well presented. Although the originality of this study is not so high, this article may provide useful information for the readers to understand the current status of HBV and HCV infections in Gabon, and will help public health organizations to develop effective countermeasures for elimination of chronic viral hepatitis in this region. 

Author Response

[Reviewer 1]

In this manuscript, the authors investigated the current status of hepatitis B and C viruses (HBV and HCV) infections and the genetic diversity of the viruses in Gabon, Central Africa. 22 HBV and 9 HCV infections detected in 2,047 patients with febrile illness. The authors found that the subgenotype A1 HBV was identified for the first time in Gabon with an insertion generating a frameshift to create an X-preC/C fusion protein, that most of the detected HCVs belonged to the “Gabon-specific” HCV subtype 4e, and that the HCV-4e strains possessed resistance-associated substitutions similar to those of other HCV-4 strains. So The authors concluded that the results of this study provide a better understanding of the current situation of HBV and HCV infections in Central Africa, and will help public health organizations to develop effective countermeasures for elimination of chronic viral hepatitis in this region. 

This is a longitudinal surveillance to investigate the current status of HBV and HCV infections in Gabon. The study was well conducted, and the results was well presented. Although the originality of this study is not so high, this article may provide useful information for the readers to understand the current status of HBV and HCV infections in Gabon, and will help public health organizations to develop effective countermeasures for elimination of chronic viral hepatitis in this region. 

Answer:

Thank you very much for the reviewer’s positive comment. We would like to conduct hepatitis virus surveillance continuously in Gabon.

Reviewer 2 Report

In this current study, Abe et al. nicely presents the surveillance data of HBV and HCV in Gabon. They use longitudinal surveillance approach and evaluate 2047 patients. It is an important study as the reduced efficacy of DAAs is most often related to polymorphisms of the subtypes.

Overall it is well-written paper with a comprehensive discussion part.

General Comments:

1-      Have you performed any structural and/or functional analysis to identify the possible roles or targets/interacting molecules of the novel fusion protein of the proteins X and preC/C? What is your hypothesis on the roles of this fusion protein regarding the resistance or viral replication/persistence?

2-      Regarding the resistance-associated substitutions for HCV-4 genotypes: substitutions at NS5A E62 and Q30 positions result in more variable aa between the different HCV-4 genotypes. Any known reasons or what is your hypothesis on that? Can you elaborate and discuss further.

Author Response

[Reviewer 2]

In this current study, Abe et al. nicely presents the surveillance data of HBV and HCV in Gabon. They use longitudinal surveillance approach and evaluate 2047 patients. It is an important study as the reduced efficacy of DAAs is most often related to polymorphisms of the subtypes.

Overall it is well-written paper with a comprehensive discussion part.

Answer:

Thank you very much for the reviewer’s positive evaluation of the manuscript.

General Comments:

Comment 1.

Have you performed any structural and/or functional analysis to identify the possible roles or targets/interacting molecules of the novel fusion protein of the proteins X and preC/C? What is your hypothesis on the roles of this fusion protein regarding the resistance or viral replication/persistence?

Answer:

We did not perform a structural and functional analysis for the fusion protein. However, previous studies provided us important insights into the function of the X-preC/C fusion protein. Hong et al. recently showed that the N-terminal signal peptide of the immature precore protein (p25) translated from the precore mRNA should be mostly removed by the signal peptidase in the ER lumen at the amino acid position of -11/-10, leading to the production of HBeAg after further processing at its C-terminal domain (Hong et al., Virology, 2023, doi: 10.1128/mbio.03501-22). This means that HBeAg might be normal even in the fusion protein due to the post-translational cleavage of HBeAg from the fusion protein.

Whereas, X protein (HBx) possibly possessed several extra amino acid residues corresponding to -25 to -11 of the precore protein at its C-terminus even after the cleavage of HBeAg from the fusion protein. Prieto et al. and Hernández et al. recently showed that HBx localization was unaffected even though they fused GFP to the C-terminus of HBx (Prieto et al., Molecules, 2021, doi: 10.3390/molecules26051254; Hernández et al., Biomedicines, 2021, doi: 10.3390/biomedicines9111701). Therefore, several extra residues at the C-terminus would lead to a minimal effect on the HBx function. As the function of HBx is essential to initiate and maintain HBV replication after infection (Lucifora et al., J Hepatol, 2011, doi: 10.1016/j.jhep.2011.02.015), HBx would function appropriately after processing from the fusion protein in the sample D0232.

   We added the above-mentioned idea to the Discussion section (page 10, lines 338-350).

Comment 2.

Regarding the resistance-associated substitutions for HCV-4 genotypes: substitutions at NS5A E62 and Q30 positions result in more variable aa between the different HCV-4 genotypes. Any known reasons or what is your hypothesis on that? Can you elaborate and discuss further.

Answer:

   There are several previous sophisticated studies that investigated the efficacy of DAA against wide range of HCV genotypes from genotype 1 to 7 (Sarrazin et al., Gastroenterol, 2016, doi: 10.1053/j.gastro.2016.06.002; Gottwein et al., Gastroenterol, 2017, doi: 10.1053/j.gastro.2017.12.015; Nguyen et al., J Hepatol, 2020, doi: 10.1016/j.jhep.2020.05.029). These studies revealed that the amino acid residues, Q30, L31, and Y93, in the NS5A region were highly divergent among genotypes, and that specific patterns of substitutions of these residues were associated with high-level resistance to DAA, such as Q30H/R/S, L31M/F, and Y93H/C/N/S. The E62 position was also divergent among genotypes, although substitutions in this position were weakly associated with resistance. Thus, the amino acid residues highly related with resistance were known to be divergent not only within genotypes but also among genotypes. Especially, a number of HCV-4 strains possessed both Q30R/S and L31M substitutions that might cause high-level resistance. It is required to validate the DAA efficacy in Central Africa where HCV-4 strains are highly prevalent.

   We added this paragraph to the Discussion section (page 11, lines 371-383) to show our idea on RASs Q30 and E62 in the NS5A region of HCV-4.

Reviewer 3 Report

The manuscript by Haruka Abe and coauthors provides a novel and interesting data on the HBV and HCV diversity on Gabon. Data on hepatitis viruses genotypes distribution and viral evolution patterns in Central Africa are scarce. The study is performed accurately and data are presented in clear way. However, the study is based on a very limited number of viral sequences (22 HBV and 9 HCV sequences), which is not enough to assess the true HBV and HCV diversity in the region. Moreover, the title of the manuscript promises the analysis of HBV and HCV evolution patterns, but no comprehensive evolution analysis has been performed in this study. The mentioned in the manuscript title “continuous genetic surveillance” that yielded only 22 HBV and 9 HCV sequences in a highly endemic region seems to not have achieved the goal of genetic surveillance. Thus, the title of the paper should be changed to reflect the main findings of the study.

Table 2. Please indicate the number of sequences of each HCV-4 subtype that were used for RAS analysis. If RASs data in Table 4 are based on single sequence of each HCV-4 subtype, please indicate the principle of choosing a sequence.

Lines 334-335. Current study has not been designed as a cross-sectional study to provide a reliable information on the HBV and HCV prevalence or provide comprehensive data on viral genetic diversity. The screening of febrile patients for HBV and HCV seems to be a convenience sampling. Thus, this statement should be softened. Moreover, authors should discuss a relatively low HBV and HCV detection rates in studied cohort compared to previously reported data mentioned in Introduction (>7% for HBV, and >10% for HCV).

There are no references to supplementary materials (tables S1 and S2) in the main text.

Authors should discuss the limitations of their study, that include the limited numbers of sequences and the studied cohort of population that did not include any patients with known liver disease.

Author Response

[Reviewer 3]

Comment 1.

The manuscript by Haruka Abe and coauthors provides a novel and interesting data on the HBV and HCV diversity on Gabon. Data on hepatitis viruses genotypes distribution and viral evolution patterns in Central Africa are scarce. The study is performed accurately and data are presented in clear way. However, the study is based on a very limited number of viral sequences (22 HBV and 9 HCV sequences), which is not enough to assess the true HBV and HCV diversity in the region. Moreover, the title of the manuscript promises the analysis of HBV and HCV evolution patterns, but no comprehensive evolution analysis has been performed in this study. The mentioned in the manuscript title “continuous genetic surveillance” that yielded only 22 HBV and 9 HCV sequences in a highly endemic region seems to not have achieved the goal of genetic surveillance. Thus, the title of the paper should be changed to reflect the main findings of the study.

Answer:

Thank you very much for an important suggestion on the title. Concerning the word “evolution”, we detected two HBV strains simultaneously in one participant and defined it as an ongoing evolution state [ref. 28 in the text]. As we detected a similar mutation at the same nucleotide position in an HBV strain in this study, we used the word “evolution” in the title. This is, however, applicable for HBV but not for HCV in this study. Therefore, we removed the word “evolution” from the title. As for genetic surveillance or diversity, several past HBV/HCV studies conducted in Gabon used the word “genetic diversity” in the title, although they included 13~15 HBV and 5 HCV strains [refs. 23, 24, and Ndong-Atome et al., BMC Infect Dis, 2008, doi: 10.1186/1471-2334-8-82]. According to these past studies, it might be possible to discuss diversity of HBV/HCV in this study.

   Thus, with the reviewer’s important suggestions, we changed the title to “Genetic diversity of hepatitis B and C viruses revealed by continuous surveillance from 2015 to 2021 in Gabon, Central Africa”.

Comment 2.

Table 2. Please indicate the number of sequences of each HCV-4 subtype that were used for RAS analysis. If RASs data in Table 4 are based on single sequence of each HCV-4 subtype, please indicate the principle of choosing a sequence.

Answer:

HCV-4 strains are prevalent mainly in Africa, but most of previous studies have used short PCR fragments of the NS5B region to determine subtypes of HCV. There are very limited number of reports that sequenced complete/nearly complete genome of HCV-4, for example, Koletzki et al., Arch Virol, 2009 (doi: 10.1007/s00705-008-0270-z) for HCV-4b and Hmaied et al., J Gen Virol, 2007 (doi: 10.1099/vir.0.83151-0) for HCV-4f. Fortunately, one previous informative study reported complete genome sequences of a number of HCV-4 genotype, although only one strain was sequenced for each subtype (Li et al., J Gen Virol, 2009, doi: 10.1099/vir.0.010330-0). To date, only one or two complete/nearly complete sequences have been available for most of HCV-4 subtypes, except for HCV-4a, 4d, 4f, 4k.

Therefore, we selected one sequence from each HCV-4 subtype to investigate RASs, with criteria that the selected strain had the smallest number of ambiguous nucleotides (N or mixed nucleotides) within each subtype and that it had a longer sequence than others within each subtype.

   We added the explanation of our criteria in the Materials and Methods section (Page 4, 2.5. Whole-genome sequencing).

Comment 3.

Lines 334-335. Current study has not been designed as a cross-sectional study to provide a reliable information on the HBV and HCV prevalence or provide comprehensive data on viral genetic diversity. The screening of febrile patients for HBV and HCV seems to be a convenience sampling. Thus, this statement should be softened. Moreover, authors should discuss a relatively low HBV and HCV detection rates in studied cohort compared to previously reported data mentioned in Introduction (>7% for HBV, and >10% for HCV).

Answer:

Thank you very much for the reviewer’s important comment. Previous serological studies reported the prevalence of HBV and HCV were >7% and >10%, respectively, as written in Introduction. According to the previous report from Gabon (Makuwa et al., J Med Virol, 2006; Ndong-Atome et al., J Med Virol, 2008), the positivity for anti-HBc IgG and anti-HCV IgG was clearly increased with the years of age. Whereas, mean years of age of our samples was approximately 12.9 years (Table 1), indicating a possibility that the sampling population was relatively young and might show a lower prevalence rate for HBV and HCV. In addition, HCV RNA was detected by nested RT-PCR in samples fewer than half the number of ELISA-positive ones in the previous study in Gabon (Ndong-Atome et al., J Clin Virol, 2009). Our surveillance was based on qPCR but not on antibody or antigen detection, possibly generating a difference in prevalence between previous studies and the current study.

   According to the reviewer’s suggestion, we added one paragraph to Discussion on above-mentioned topic in lines 327-337 (page 11). Moreover, we removed a sentence in lines 334-335 where the reviewer showed a concern.

Comment 4.

There are no references to supplementary materials (tables S1 and S2) in the main text.

Answer:

   Table S1 shows primer and probe sequences used in this study and is cited in line 159 in the Materials and Methods section. Table S2 shows the result of BLAST analysis of HBV strains and is cited in lines 222 and 224 in the Result section.

Comment 5.

Authors should discuss the limitations of their study, that include the limited numbers of sequences and the studied cohort of population that did not include any patients with known liver disease.

Answer:

   The limitation of this study is that our analysis was based on passive surveillance. qPCR tests for HBV and HCV were implemented in patients who visited local hospitals in Gabon, which might induce a bias in the selection of a target population. Actually, the mean years of age of all participants was 12.9 years, indicating that a large number of participants were children aged less than 12 years. Moreover, previous studies conducted in Gabon were serological screenings for HBV and HCV, and PCR was used only for antibody/antigen-positive samples, although our surveillance performed qPCR to detect HBV and HCV [23-26]. For better understanding of the prevalence, it might be required to combine qPCR and antibody/antigen detection in HBV and HCV surveillance.

   We added the above-mentioned limitations in Discussion (page 12, lines 384-392).

Round 2

Reviewer 3 Report

Dear Authors, 

Thank you for uploading the modified manuscript. All comments are addressed in revised manuscript. The paper can be published in present form.